# Competences to Address SDGs in Higher Education—A Reflection on the Equilibrium between Systemic and Personal Approaches to Achieve Transformative Action

**Jana Dlouhá [1],\*, Raquel Heras [2] 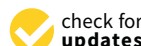, Ingrid Mulà [2], Francisca Perez Salgado [3] and Laura Henderson [1]**

[1] Environment Centre, Charles University, 16200 Praha, Czech Republic
[2] Institute of Educational Research, Universitat de Girona, 17004 Girona, Spain
[3] Department of Science, Faculty of Management, Science and Technology, Open University of the Netherlands, 6401 DL Heerlen, The Netherlands
\* Correspondence: jana.dlouha@czp.cuni.cz; Tel.: +420-604-670-025

**Abstract:** Competence-oriented teaching that leads to the sustainable transformation of both the individual and society requires a holistic learning process that addresses the cognitive, socio-emotional and behavioural domains of learning in a balanced way. This article questions whether a personal dimension of competences (addressing the individual's values, attitudes, and lived experiences) is relevant for higher education in addition to the systemic dimension (learning objectives emphasizing cognitive processes). A theoretical concept for analysing competence frameworks from this point of view was developed in a multi-step qualitative research process: two existing competence frameworks underpinning international ESD policies were compared and, based on the findings, an analytical tool to analyse competence dimensions was drafted as a two-dimensional matrix. This tool was tested on competence frameworks reported in the literature and on examples from practice in confrontation with related academic discussion. The analysis of sustainability competences with this tool illustrates the transformative dimension on a scale from holistic thinking through future orientation to achieving transformation, and the normative dimension that indicates the complementarity of the personal and systemic character of competences. The analysed competence frameworks include competences more or less evenly distributed in both dimensions; the competences in the socio-emotional learning domain were often associated with envisioning change and achieving sustainable transformation. As anticipating the future in an active way is relevant for sustainability-oriented HE programs, not only should this dimension of competences be afforded greater consideration, but pedagogies addressing the personal level should also be further investigated and implemented in HE.

**Keywords:** higher education; ESD; competences; future thinking; sustainability; transformation; learning dimensions; socio-emotional domain

## 1. Introduction

Due to its complex and normative nature, sustainability requires the involvement of different stakeholders, each with their specific interests, experiences and tools, to achieve their distinct but related (sustainability-oriented) goals. One of these significant stakeholder groups are scientists who cooperate across disciplines to find new solutions to sustainability-related problems. Educators who prime new generations to face the challenges of current unsustainable development have a role in shaping the future as well. Both science and education have made changes in practices, perspectives,

scientific (disciplinary) or educational methods and policy frameworks in the face of pressing global issues. They are also closely interconnected, particularly in the field of higher education (HE), where changes in science directly influence changes in education and demands on early career researchers in terms of their skills and competences.

This paper seeks to develop an operational framework in order to identify common principles and trends in sustainable development, sustainability science and education for sustainable development. Sustainable development (SD) is a normative concept related to the explicit agreement on limiting human activities to create a safe space for operating within planetary boundaries [1]. From a policy point of view, the Sustainable Development Goals (SDGs) have been accepted internationally as a proxy for SD, to be implemented in specific, diverse contexts. These goals were set within a broad, international debate among different social groups under the initiative "The Future We Want". The process of SDG formulation was described as collective envisioning and participative debate towards making these desired future visions tangible. The implicit assumption is that these goals will have a transformative effect on our societies. The concept of sustainability science offers a normative perspective in scientific research, making it possible to find sustainability solutions in many fields of practice while having a transformative impact on our societies [2]. Drawing on recent discussion on education for sustainable development (ESD), we specifically pay attention to the concept of competences as formulated by UNESCO [3], how these are articulated through all the SDGs and to what degree competences are applicable to, and learnable in a HE context in particular.

## 1.1. Sustainability-Oriented, Competence-Based Education

The concept of competences integrates knowledge with skills and attitudes (While the *Bloom's taxonomy of learning objectives* [4], updated by [5,6] was designed for processing abstract rather than practical knowledge (and hence addresses affective and psychomotor domains separately), the concept of competences plays an integrative role: combining knowledge (theoretical background), skills (methodological capabilities within a discipline or area of practice), and attitudes (values that provide a moral compass for behaviour and decision making). Values and attitudes shape everyday decisions and thus behaviour in sustainability context [7]), revealing the dynamic, context-specific character of learning that comes into play whenever an individual applies knowledge acquired (theoretically or practically) to achieve an observable result (an action or product). This dynamic perspective represents a substantial shift in the understanding of cognitive and non-cognitive/affective processes in learning and research. Competences underpin not only the instrumental processes of knowledge transfer and appropriation, but also activation in goal-oriented, ethically grounded actions in practical situations [8]. In the field of sustainable development, the integrative character of competences is especially important. SD draws on cognitive, as well as normative and motivational factors as an important pre-requisite for making informed decisions, planning and making interventions [9] (p. 757). Due to the character of competences, their development cannot be based on the transmission of knowledge but requires changing teaching/learning methods: they must be *learned* (through practice) rather than *taught* (theoretically) [10–12].

In this context, *sustainability competences* are indispensable for critical reflection on unsustainable patterns of behaviour and the reorientation of human activities towards sustainability ([13]. Sustainability competences are essential for building capacities which enable individuals to critically review prevailing values, policies and practices, and empowering them to make decisions and act for change [14] (p. 196). Thus, sustainability competences have an emancipatory and transformative impact [15]. Michelsen and Adomssent [16] stress the formative role of competences that is inherent in the German concept of 'Gestaltungscompetenz', which is described as 'a forward-looking ability to modify and shape the future of our society in terms of sustainable development, through active participation' [17,18]. In this article, we concentrate on the competences which have an emphasis on values, and that are articulated through the ability to plan and make decisions, reflect/reframe/transform current patterns, design/shape/envision, and act, all in the service of reaching sustainability-related

goals. We consider competences to be both a prerequisite and an outcome of learning that is essential for sustainability literacy, i.e., using knowledge to describe, analyse, and/or address sustainability issues in different contexts, but also underpin normatively-guided (sustainably relevant) informed decision making and actions in complex situations [3]. Competences are essential for applying available knowledge and expertise to address particular sustainability issues at a *systemic level* (in the social sphere and in policy-making), but they also work at a *personal level* guiding individual choices and lifestyle formation [19] (p. 354).

This dual systemic and personal perspective may require different approaches for competence development. In this respect, Barth and Michelsen [20] (p. 110) refer to the interaction of cognitive and non-cognitive/affective components of competences. In the first case, competence development is associated with increasing mental complexity and building new mental models in response to the complex demands of the contemporary world. The development of non-cognitive components on the other hand is explained through an interiorization process, i.e., value production, reproduction, communication and acquisition. The interplay of these two approaches facilitates a transformative influence on an individual in a non-manipulative way, through the development of his/her capabilities to be active in the transformation processes [20] (p. 110–112). If appropriately developed, both cognitive and non-cognitive/affective levels of competences then help learners to hone and apply their sustainability expertise in practice, and also empower and motivate them to become active citizens, capable of critically engaging in the collective process of creating a sustainable future.

## 1.2. Role of Scientific and Educational Discourses in Higher Education

While the traditional model of science is value-free, sustainability science is expected to provide a framework to address uncertain problems with value-based solutions [21,22]. Sustainability research should generate evidence-supported, actionable knowledge that is based not only on an understanding of the problem, but also guided by a sustainability-inspired vision, that is applied to delineate transition and intervention strategies. Thus *normative knowledge* provides insight into the problem from various time perspectives, past, present and future (and values are thus integrated in its assessment and visioning methods), whereas *instructional knowledge* is applied consequently, helping to resolve the problem or achieve the vision (while the values implicit in societal versus technical contexts might differ). Both types of knowledge are used in intervention research to support change and simultaneously reflect on it, or to address more complex, 'wicked' problems with a transition management and governance approach [22] (p. 33–35). In this broader research framework, *sustainability competences* have been defined as playing a vital role within the interdisciplinary higher education programs aiming at societal transformation [23].

The personal versus systemic perspective is also discussed in social sciences as the nomothetic vs. idiographic dichotomy that underpins the use of different scientific approaches and research methods. The nomothetic approach to knowledge is associated with the tendency to generalize, focusing on objective phenomena, while idiographic approach is related to the understanding of unique, often culturally specific or subjective issues (nomothetic methods are often quantitative, while idiographic are qualitative) [24]. Knowledge delivered at the HE level is mainly of a nomothetic, systemic nature; in this context, the demand for competence-based, sustainability-oriented learning is often translated instrumentally, paying greater attention to skill acquisition while bypassing the examination of and reflection on values and attitudes [11]. However, pedagogical discourse is concerned with the development of the individual as a whole—thus being in principle value-based, with the ethical dimension of human development at the centre of attention and developing particularly socio-emotional and behavioural competences through relevant learning processes in practice.

That HE tends to side-line this normative principle is documented by Lozano [25], who showed that the academic debate on pedagogical approaches addressing socio-emotional and behavioural aspects of sustainability competences is held separately from the competence debate. Conversely, Barth and Michelsen [20] show that these two discourses have the potential for mutual enrichment:

new pedagogical approaches (based on 'problem-based learning, social learning, situated learning, and social-constructivist approaches to learning' (ibid, p. 107) can rapidly evolve to support competence development if systematically related to sustainability challenges such as uncertainty, complexity and interdisciplinarity. Shephard, Rieckmann and Barth [26] (p. 6) show that 'a particular conceptualisation of pedagogy' is necessary to develop decision-making capacities and competences associated with freedom of choice. Albareda-Tiana et al. [10] suggest that anticipating and preparing for future sustainability challenges, integrative thinking and participation as core elements of ESD require application of, and change in, teaching methods and approaches towards competence-based teaching and learning; similarly, Biasutti [27,28] document the need to rethink teaching in the ESD context and to connect the development of values, skills and behaviour with relevant didactic strategies. As values play a crucial role in sustainability, this research aims to provide guidance for recognizing/assessing this normative aspect in teaching/learning processes and its outcome competences.

### 1.3. Competences in HE Teaching/Learning

Competence-oriented curricula are not easy for educators at all levels of education: the HE context is no exception. Competences that manifest themselves in action are usually learned through action and, even though they represent a distinct learning outcome, they are described as 'learnable, not teachable', hence presenting a challenge to training them in a learning environment [19] (p. 356). The difficulties manifested in defining competences or the different understandings depending on different contexts [26,29] make it difficult for teachers to situate themselves and their work in the bigger picture. Moreover, the multifaceted nature of the concept hinders its assessment and it can be difficult to measure if behavioural change has truly been manifested [30].

However, international policies in HE support competence-oriented concepts of learning: the Bologna Process entailed a profound restructuring of the European higher education system in this respect. Neave and Veiga [31] identify a widespread perception of the fact that substantial changes have been achieved concerning teaching methods and student participation in learning activities, and the promotion of flexible learning paths. According to Mateo et al. [32], there is 'a general trend for curriculum guidelines to shift from being content-oriented to being learning-oriented curricula. The new pedagogical models focus on learning acquired through personal work, self-regulation and on the establishment of ideal conditions for achieving the educational goals.' (ibid., p. 435). Competence acquisition occurs in physical, social and cultural contexts and requires mediation between knowledge acquired and the contexts in which learning occurs. Thus, the Bologna Process is understood as constituting pedagogical reform and a chance to move ahead with student-centred learning methodologies and the development of competences grounded in best practices. Since then, many higher education reports and curricula have been transformed to competence-based curricula, in line with what is desirable to progress towards ESD [33].

ESD's specific characteristic is the commitment to transforming learners' educational experiences so that they can evaluate, question and challenge personal lifestyles and professional responsibilities, as well as explore their motivations to act for a more sustainable world [34]. ESD is concerned with thematic content associated with sustainable development but is more interested in challenging traditional pedagogical strategies focused on the transmission of expert knowledge. Instead, it promotes learning spaces for people to develop key competences to respond to complex and uncertain future scenarios [35–38]. ESD is thus associated with a wide variety of pedagogical principles that support learner-centred approaches, such as critical and creative thinking, participation and participatory learning, and action learning. It also supports other unique and less common pedagogical principles such as futures and systems thinking (see [38–40]). Competences play a pivotal role in all these approaches, and are at the centre of interest of key educational actors as documented in participatory research conducted for developing a commonly agreed competence framework [41]. As these principles must also be applied in sustainability-oriented HE, it is crucial to explore all the dimensions of competences, including behavioural and socio-emotional dimensions, in this context.

## 2. Research Questions

While education is holistic in principle, concerned with the coherent development of human beings within their society, the traditional science (as contrasted to sustainability science) is reductionist, specifically focused on the subject of exploration. The development of competences could also be viewed through the different lenses of these two perspectives: the former working contextually, with humans and their desires, attitudes, and consequently shared norms and values, the latter universal, objective, delivering solutions for application across the board.

Barth and Rieckmann [42] have found that teaching and learning approaches are a frequent theme in higher education for sustainable development research (i.e., the adequacy and applicability of these approaches for developing sustainability competences), but little is known about competences as learning outcomes. This is the point of departure for this research that is focused on sustainability competences that support transformation at the personal and societal level, where value-based decisions play a key role. We hypothesize that in the context of what has been said above, the normative and *personal dimension* of sustainability competences is crucially important, alongside the instrumental and *systemic dimension*, but that this dimension is not sufficiently covered in the existing competence frameworks, especially at the HE level. Thus, our questions are:

1.  Is it possible to distinguish between the instrumental (systemic) and normative (personal) dimensions of competences?
2.  If so, do the existing competence frameworks (specifically the SDGs competence framework) work with these dimensions, and how?
3.  Are all of these dimensions applied in the practice of HESD? If so, is there any reference to pedagogies needed to address those dimensions considered relevant?

## 3. Methodology

To gain theoretical insight into the problem, qualitative methods based on grounded theory [43,44] enable working with emerging phenomena, concepts and theories. These methods were adopted to analyse and compare two overarching competence frameworks and reflect on practical ESD endeavours. However, in the sustainability field, the goal is often not primarily to find new concepts but instead to help reframe existing concepts so that they comply with normatively set goals. To this end, a conceptual framework analysis method [45]) was applied—the authors used different sources of information, compared and interpreted them in several steps with the intention of gradually developing a new perspective on the problem [46].

An initial understanding of the role of competences in ESD was developed within previous research where the UNECE framework for ESD competences [47] was used for comprehensive analysis of the higher education environment in 53 institutions from 33 European countries [34,48,49]. The role of competences is significant in transition towards ESD at the HE level, especially its more advanced stages, confirming the intricacy and innovative nature of the competence concept [50]. In this research, the role of competence dimensions (normative and transformative), as highlighted by the UNECE framework [47] was further explored in more recent documents where it appeared to be present somewhat implicitly. As attention paid to the competence dimensions should not be lost in the policy and academic debate, the authors made an effort to develop an analytical tool to reflect their presence in different contexts. The tool was drafted as a simple generalization of the UNECE framework [47] that would allow the categorization of the competences. This step was instrumental in answering the first research question: to systemically distinguish cognitive, social, emotional (affective), and conative/behavioural domains of learning; and the holistic, future-oriented and transformative character of learning processes.

To justify the validity and applicability of the draft tool and its appropriateness for analysis, a theoretical open coding method [51] was used. The UNECE framework [47] provided theoretically grounded categories (These categories closely follow the headlines in the UNECE [47] (p. 6) matrix:

on one coordinate, expressed as learning domains (learning to know; learning to do; learning to live together; learning to be), and on the other, as dimensions of learning (holistic approach; envisioning change; achieving transformation)) with which the competences under the UNESCO framework [3] were coded. Codes outlined from the theoretically grounded UNECE framework [47] were used to identify relevant elements of competence description from the UNESCO framework [3]. (For example, the 'learning to know' and 'holistic thinking' codes matched with the following characteristics of the Systems thinking (ST) competency from the UNESCO framework [3]: 'to recognize and understand relationships; to analyse complex systems; to think of how systems are embedded within different domains and different scales'. However, some aspects of this competency also fit with 'learning to do' and 'holistic approach' codes in the UNECE [47] matrix because ST characteristics: 'to deal with uncertainty' already includes an action. In this way, also other key sustainability competences were compartmentalized into the UNECE [47] matrix). The categories were designed to uncover in particular the personal and systemic character of competences and their transformative nature. The analysis of the key sustainability competences outlined in the UNESCO framework [3] showed that all eight of them fall into more than one of the categories. As both of the compared frameworks are similarly justified in scientific and policy discourse (playing a role in international strategies), and thus they are indicative of the "state of the art" of ESD debate, this exercise was sought to be potentially productive with regards to generalizations. The results of this step were consequently discussed in reference to other concepts and frameworks that appear in academic debate—thus addressing the second research question, pointing out in particular the role of the normative and transformative dimensions within the UNESCO competence framework [3].

However, due to ESD's normative nature, both policy and scientific concepts related to ESD are rather speculative, and there is a need for practical feedback to justify their applicability. In this research, theoretical discussion in the previous steps was further combined with the case study method [52,53], enabling critical reflection on practice. We used the draft tool to analyse some of the case studies, thus testing its applicability in practice, and illustrating pedagogical approaches that are used for competence-based teaching at the HE level. The relevance of the competence dimensions for HESD, and related need for appropriate pedagogies, was discussed in the final stage to answer the third research question.

To summarize, drawing on the UNECE ([47] matrix containing specific competences of educators for ESD, a general analytical tool was developed with the aim of making it possible to distinguish between the two competence dimensions—normative and transformative—in all the competence systems under exploration. The research questions were addressed at a theoretical level and in different contexts of practice.

These steps taken in the research are described in Figure 1.

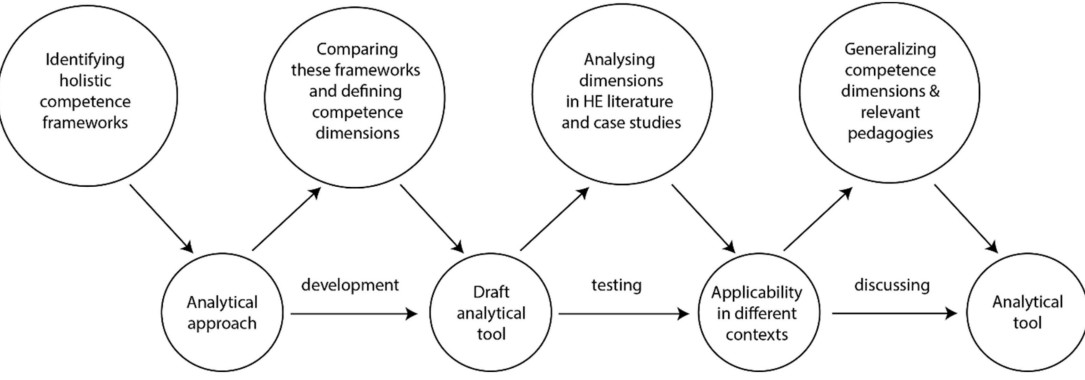

**Figure 1.** Progress of the research.

## 4. Developing Analytical Tools

Each of the research steps (Figure 1) used the concepts and methods described below.

### 4.1. Search for Holistic Competence Frameworks

The concept of competence is multi-dimensional and few competence frameworks specify its learning domains; those identified in this research are compared in Table 1.

The cognitive domain is described in a similar way by all of the authors; the other domains differ (and overlap) to a certain extent. Bloom's [4] taxonomy (and subsequently revised by Anderson and Krathwohl [5]) are not competence oriented—these authors strive to separate domains rather than integrating them. Consequently, also the learning processes applied in scientific training may stress cognitive, and to a large degree disregard the conative and especially affective domains despite the fact that values, which are basis of decision making, become the guiding principle of action, and are embedded in sustainability science.

**Table 1.** Comparison of learning domains outlined by the cited authors (cf. also Dlouhá, 2009a, 2009b).

| Authors/Domain | Systemic Approach → Growing Importance of Personal Approach | | | |
|---|---|---|---|---|
| [4,5,54] | Cognitive | Psychomotor → Conative [1] | Affective | |
| [47,55] | Learning to know | Learning to do | Learning to be together | Learning to be |
| [3] | Cognitive | Behavioural | Socio-emotional | |

[1] In Bloom's taxonomy [4], the psychomotor domain reflects mainly manual skills, while in the revised model, the conative domain is associated with action [54].

### 4.2. Defining Competence Dimensions

Learning domains and dimensions outlined in the UNECE framework [47] provide an insight into the role of competences in learning processes; this role is, however, not explicitly expressed in the latter frameworks. This stimulated our interest in comparing these frameworks to answer the second research question: are these dimensions still present in the definition of sustainability competences?

To test its analytical character, the UNECE [47] table was used to develop categories that allow the positioning of competences or their elements from other explored frameworks on a scale within the two dimensions, which we further describe as transformative (relevant for the sustainability change process) and normative (illustrating the divergence between systemic and personal dimensions)—see Figure 2.

Categories in the transformative dimension indicate progress towards the desired future while categories in the normative dimension allow us to distinguish between learning domains. In the transformative dimension, a *holistic* approach and *future*-oriented thinking are pre-requisites of *transformation*; in the normative dimension, *learning to know* is the most instrumental level (achievable without considering values, with ostensibly value-free *instructional* knowledge) and *learning to be* occurs at a greater depth of the normative dimension (values and *normative* knowledge are most important). It is expected that becoming sustainability-literate requires an individual to develop competences in both dimensions in a balanced way. Positioning competences within these dimensions should depict the distinction between their personal and systemic character.

To examine this assumption, the categories from the UNECE framework [47] have been used as codes to analyse the latter UNESCO competence framework [3] (Where the general description of learning dimensions mentioned above was not sufficient, more detailed examples of competences from the original UNECE [47] competence matrix were used for justification of the codes. For example, when the Learning to live together/Envisioning change category is described as the ability to "facilitate the emergence of new worldviews that address sustainable development"; and to "encourage negotiation of alternative futures" in the UNECE [47] document, this has been translated as the "ability to collectively develop … " (key competence S), to "understand, relate to and be sensitive to others (empathic leadership)" (key competence C), and to "reflect on one's own role in the local community and (global) society" (key competence SA)—see Table 2). The elements of key sustainability competences from this framework that possessed similar codes were included into the relevant categories of the UNECE [47] matrix, as apparent from Table 2.

**Table 2.** The competence matrix used to analyse the key competencies for sustainability from UNESCO [3] p. 10). These sustainability competences in this table are abbreviated as follows: Systems thinking competency = ST; Collaboration competency = C; Anticipatory competency = A; Critical thinking competency = CT; Normative competency = N; Self-awareness competency = SA; Strategic competency = S; Integrated problem-solving competency = IPS.

| | Holistic | Envisioning | Transformative |
|---|---|---|---|
| Learning to know Cognitive domain | ST (recognize and understand relationships; to analyse complex systems; to think of how systems are embedded within different domains and different scales … ) CT (question … practices and opinions … in a context of conflicts of interests and trade-offs, uncertain knowledge and contradictions) | A (understand and evaluate multiple futures) N (understand and reflect … in a context of conflicts of interests and trade-offs, uncertain knowledge and contradictions) | S (collectively develop and implement innovative actions that further sustainability at the local level and further afield) IPS (apply different problem-solving frameworks to complex sustainability problems) |
| Learning to do Conative domain | ST (deal with uncertainty) | A (assess the consequences of actions) S (develop and implement innovative actions) | A (deal with risks and changes) IPS (develop viable, inclusive and equitable solution options that promote sustainable development) |
| Learning to live together Social domain | CT (question norms, … and opinions) C (learn from others; to understand and respect the needs, perspectives and actions of others (empathy)) | N (negotiate sustainability values, principles, goals, and targets) S (collectively develop … ) C (understand, relate to and be sensitive to others (empathic leadership)) SA (reflect on one's own role in the local community and (global) society) | S (collectively develop and implement) C (deal with conflicts in a group; facilitate collaborative and participatory problem solving) IPS (integrate all of the key sustainability competences) |
| Learning to be Emotional domain | SA (deal with one's feelings and desires) | A (create one's own visions for the future) N (understand and reflect on the norms and values that underlie one's actions) CT (reflect on own one's values, perceptions and actions) SA (continually evaluate and further motivate one's actions) | CT (take a position in the sustainability discourse) SA (deal with one's feelings and desires) |

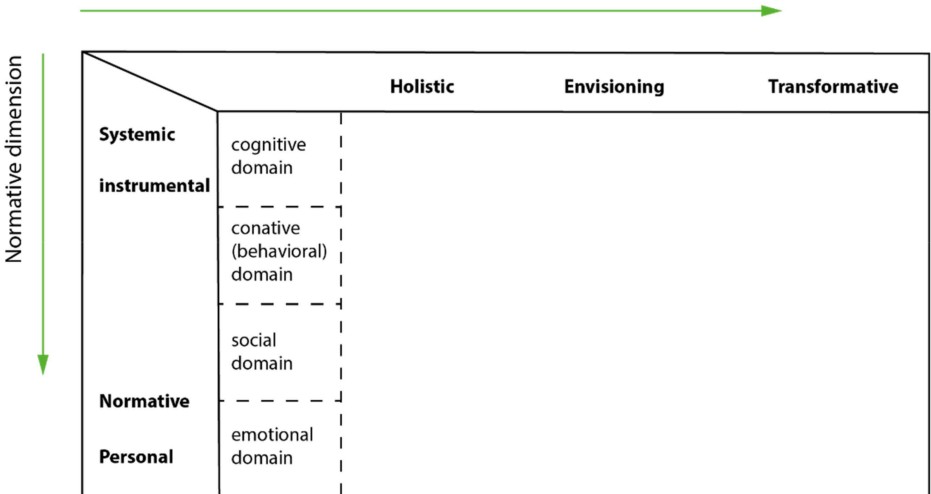

**Figure 2.** Dimensions in achieving sustainability competences.

## 5. The Personal (Versus Systemic) Dimension and Relevant Pedagogies in Higher Education

Competence dimensions are further addressed in reference to the relevant literature with the goal of reflecting on whether they provide an insight into the discussed frameworks or even reframe them, and also to discuss the role of pedagogies.

The competence directly associated with future is the 'Gestaltungskompetenz' or 'shaping competence' that was defined by De Haan [17] (pp. 22–25) as a framework competence in sustainability, necessary for acting and solving problems. This competency approach creates space for multiple possible futures that can be actively shaped. It encompasses several sub-competences, such as foresight thinking, interdisciplinary work, 'reflection on individual and cultural models', participatory skills and others [17] (p. 24). At a personal level, it is based on the 'capacity for empathy, compassion and solidarity' and 'competence in self-motivation', while on a systemic level, competency in planning and implementation is an important component (ibid.). The development of this competence requires specific teaching and learning approaches to attain 'intelligent, interconnective and application-related knowledge'. From a pedagogical point of view, a *situated learning approach* is highly relevant here, as is individualized instruction and the application of the 'learning communities' principle (ibid.), thus emphasising the normative dimension (socio-emotional domain) in competence development.

The personal dimension of competences plays a role also in the Responsibility, Emotional intelligence, System orientation, Future orientation, personal Involvement, Action skills + disciplinary competences (RESFIA + D) model of professional competences for sustainable development in HE developed by Roorda and van Son [56] (p. 335). According to these authors, HE's primary goal is to educate professionals [56] (p. 339), which in a sustainability context requires future-oriented competences in the cognitive domain, such as 'thinking on different time scales, recognition of non-linear processes, thinking innovatively, creatively, and out of the box' (ibid., p. 341). Of particular importance is the *Personal involvement* competence, that emphasises the professional's personal engagement, and hence consistently integrating sustainable development throughout one's professional work enabling the professional to 'work passionately towards dreams and ideals, and employ one's conscience as the ultimate yardstick' thus developing competences in behavioural domain. *Emotional intelligence* is considered to be necessary for the explicit recognition of, and respect for, one's own values and those of other people and cultures. The three learning domains (cognitive, behavioural and socio-emotional) are all present in this model which, while it is rich in holistic and envisioning competence dimensions, lacks the transformative dimension.

A value-based perspective is emphasized also by Martens, Roorda and Cörvers [57] among others, who describe a sustainably competent professional as someone who 'works and thinks on the

basis of a perspective of the future' [56] (p. 341). According to these authors, emotional intelligence and personal involvement are needed as much as (instrumental) action skills. In terms of pedagogy, new educational methodologies are required to enhance these sustainability competences, such as *problem-based learning*. The framework outlined by Lozano et al. [25] is similarly balanced in that it synthetizes sustainability competences in all dimensions, including anticipatory thinking and strategic action in the systemic dimension, while empathy and change of perspective, and personal involvement feature in the normative dimension. Envisioning the future and achieving transformation is associated with creativity [58].

An attempt to integrate both personal and systemic aspects is apparent in the *intervention competence* introduced by [59–61]. The authors describe the distinction between intervention and action: action is considered to be the habitual act one performs without critically thinking while intervention can be seen as a new act which is consciously thought over, and has its starting point in a creative and critical future-oriented thinking process located in a multi-actor setting.

## 6. Case Study Analysis

The practical relevance of all dimensions in achieving sustainability competences, and required pedagogies, are discussed in the following case studies. However, there are some methodological constraints. The work of several authors [61–63], in which they investigate sustainability professionals and alumni working in the field, suggests that conducting the process towards sustainability demands a large, diverse set of abilities, and that these should receive more attention in HE programs. However, the need for universalism in competency analysis and in the formulation of frameworks has been criticised [15], and this means that different competency frameworks (and their combinations) might be best fitted to different contexts. To show the role of the context, in the following section, the authors describe their practical experiences: a course for students of environmental sciences (The Netherlands), and a course organized within a teacher education program (Spain). The third case study is concerned with university educators' competences as explored through the cooperation of numerous higher education institutions during a three-year international project (Europe).

### 6.1. Case Study: Lived Experience of Climate Change

#### 6.1.1. Case Description

An Open Education Resources (OER) Mastertrack with the name of The Lived Experience of Climate Change (approximately 400 study hours) has been developed by a consortium of eight European higher education institutions [64]. In this consortium, an interdisciplinary group of natural scientists, social scientists and engineers cooperated. Part of this e-learning track is used in the Premaster of Environmental Sciences of the Open University of the Netherlands, and the theme 'Lived Experience of Sustainable Development' is optional for students.

The Mastertrack is an example of an educational programme on Sustainable Development Goal (SDG) 13: Climate Action. This program is innovative in that it is competence-based and that it expands the scientific knowledge base: it is integrative, transdisciplinary, and connects science with humans as individuals and citizens.

#### 6.1.2. Contents and Pedagogies Used

From a constructivist view, cultural and social aspects will influence the production of scientific knowledge. According to Foucault [65] and Haraway [66] power relations in a society are reproduced in the knowledge production, which leaves the 'less powerful' unrepresented or misrepresented. This will also apply for sustainability science, because universities are also subject to these forces [67]. The climate change discourse has been dominated for several years by 'objective', controlled technical and scientific criteria. However, a growing climate justice movement is questioning the ethics and politics of local and global power structures and is combined with action on climate change on this basis.

These concerns and ideas have led to the introduction of the people-centred concept, The Lived Experience of Climate Change [64,68,69]. Lived Experience is described as the knowledge gained and obtained by people over time through their engagement with each other and the knowledge they generate through their actions. The concept connects experiential learning to action and recognises the social conditions of knowledge production and engagement. This lived experience is influenced by a diverse set of more enduring factors, such as gender, one's social class, one's education, local and regional values.

A few characteristics of the pedagogies stand out in this Mastertrack: the curriculum contains tasks and cases based on authentic situations, to which the students can add and introduce their own problem/case/theme related to their own lived experience; they carry out tasks in project-based learning activities; the teaching style is a combination of teacher- and student-centred learning; there is a certain flexibility within the curriculum to address the themes brought in by the students, within the framework of the university; in the assessment the students demonstrate their competences and there is an emphasis on various types of performance assessments (presentations, project reports) which simulate authentic situations [59].

### 6.1.3. Competence Analysis

The Mastertrack contains, in addition to two other competencies (diagnosis and transboundary), a competency focusing on transformative action. This competency is labelled 'intervention competency' [58–60] and is defined as: "The ability to devise or propose, in a process of consultation with relevant stakeholders and actors, one or several sustainable solution(s) for a climate change issue."

This competence contains the following dimensions from the cognitive, socio-behavioural and emotional domains, based on empirical results obtained in cooperation with sustainability professionals in the Netherlands [60,70]. They are described as:

The ability to:

- combine scientific and experiential knowledge in order to create an integrated assessment of a climate change issue. Firstly, the knowledge from the natural sciences (chemistry, physics, geography, biology) is needed. Secondly, scientific knowledge of the social, political, economic, cultural and gender aspects is added. Crucial, however, is connecting one's own lived experience to this scientific knowledge;
- be aware of the importance to reach decisions or interventions; goal-oriented action;
- engage in strategic and political thinking, deliberations and actions; relate these to different perspectives of the stakeholders;
- cope with complexity;
- adopt and communicate ethical practices to the stakeholders;
- translate the stakeholder diversity into collectively produced interventions for the sustainability issue.

These dimensions interact and work in combination with each other. Intervention competency helps students to formulate and implement actions for a climate change issue, while appreciating the diversity of global and local societal aspects. It integrates the systemic (instrumental) and personal (normative) aspects, because it connects the student's lived experience with scientific knowledge. Students learn with the intervention competence to think beyond 'only' studying a problem: they connect their own lives with climate change. They also learn to work with stakeholders and to formulate ways and options for reaching decisions or interventions—this should effectively prepare them for conducting change processes in climate change issues.

*6.2. Case Study: Sustainable Schools Module in Teacher Education*

6.2.1. Case Description

The example described is an optional module of three European Credit Transfer System—ECTS (75 h) offered by the Faculty of Education and Psychology of the University of Girona, Catalonia, Spain, to 2nd-year students of the Bachelor's degree in Early Childhood Education and the Bachelor's degree in Primary School Education. In the module called Scientific and Environmental Experiences in Schools, approximately 40 students are enrolled each academic year (out of 170). The module's content is designed to empower students to work in sustainable schools and involve them in leading projects and practices related to sustainability, differentiating the module from other subjects that treat sustainability rather theoretically.

According to the IBE-UNESCO document, teacher education worldwide has undergone a process of adjustment in response to current global economic, social, cultural and political challenges. As Mulà et al. [34] show, students are also vocal and active in society but have been shaped by an education system that does not question, or offer an alternative, to the cultural norms, ambitions and practices steeped in consumption and exploitation. While universities may raise awareness of sustainability, it may not be enough to promote deeper changes. This is decisive in pre-service teacher training, where professionals of education can influence elementary education and the management of schools. For this purpose, several theoretical frameworks addressed specifically to teacher training have been developed in different universities. Two examples are the KOM-BiNE competency model [71] and the 'Professional Competencies in Education for Sustainability' model [72] —both of them include affective and interpersonal domains and offer different contexts to promote action, reflections and critical thinking, and envisioning future and different scenarios, among other competences.

6.2.2. Contents and Pedagogies Used

The module is focused on responsible consumption and biodiversity loss. Consumption is a topic close to students' daily experience, as every day we consume and, doing so, we make choices and we act. Key aspects of the learning activities in the module, besides concrete experiences of students' everyday life, are also the students' future professional practices at kindergartens and elementary schools.

Contents related to the four contexts (curricula, democratic participation, school building management, school community engagement (cf. Catalan Government) are approached with active pedagogies, such as role-playing, gamification, viewing short films, group discussion and decision making, getting to know inspiring projects and people, outdoor learning and envisioning of future alternatives. In practice, starting with a role-play activity includes the participants having to prepare a welcome party for new students at the beginning of the course, analysing the different products used at the party with regards to their production, life cycle, waste generation and management, and whether or not they are healthy. In doing so, the three dimensions of sustainability are considered in the assessment of the products' impacts. Working in small groups, students have to design a more sustainable party that is then organized at the end of the module as a goodbye party. This activity creates the opportunity to link with other activities in the module that are related to the school context, such as eco-auditing, co-creation projects with the whole school and the community about the "dreamed school" (the ideal school we want and what to do to realise it) and the impact of waste in biodiversity loss (local-global dimensions).

To encourage the reflection process and critical thinking, a reflective-based learning approach [73] evaluation tool for this module is used: the student portfolio. In the portfolio, students include some written reflections and documents and materials related to their interests or topics covered during the class. To help students with personal and deep reflection, a rubric based on Alsina et al. [74] is provided.

The module focuses on two SDGs [6], especially SDG 12, responsible Consumption and Production, and to a lesser degree SDG 15 (Life on Land). (The contents related to SDGs 12 and 15 [6] (pp. 34,

40) are: production and consumption patterns and value chains, management and use of natural resources (renewables and non-renewables), environmental and social impacts of production and consumption, food production and consumption (agriculture, food processing, dietary choices and habits, waste generation, deforestation, overconsumption of food and hunger), waste generation and management (prevention, reduction, recycling, reuse), sustainable lifestyles and diverse practices of sustainable production and consumption (labelling systems and certificates), life cycle analysis of different products; and threats to biodiversity: habitat loss, deforestation, fragmentation, invasive species and overexploitation.) The SDG 12 learning objectives have been analysed with the draft analytical tool (see Table 3).

**Table 3.** Overview of SDG 12 learning objectives cited from the UNESCO [3] (p. 34) document as used in the Sustainable Schools module of the University of Girona. Objectives highlighted in bold are applicable (in addition or exclusively) at the personal level. The numbers characterize learning objectives according to the domain (1. cognitive, 2. behavioural, 3. social-emotional), and the number of the learning objectives in this domain.

| | Holistic | Envisioning | Transformative |
|---|---|---|---|
| Cognitive domain | **1.1. The learner understands how individual lifestyle choices influence social, economic and environmental development.** 1.2. The learner understands production and consumption patterns and value chains and the interrelatedness of production and consumption (supply and demand, toxics, $CO_2$ emissions, waste generation, health, working conditions, poverty, etc.). | **1.4. The learner knows about strategies and practices of sustainable production and consumption.** 1.5. The learner understands dilemmas/trade-offs related to and system changes necessary for achieving sustainable consumption and production. | |
| Conative domain | 2.5. The learner is able to challenge cultural and societal orientations in consumption and production. | **2.4. The learner is able take on critically on their role as an active stakeholder in the market.** | **2.1. The learner is able to plan, implement and evaluate consumption-related activities using existing sustainability criteria.** |
| Social domain | | **3.1. The learner is able to communicate the need for sustainable practices in production and consumption.** | |
| Emotional domain | **3.3. The learner is able to differentiate between needs and wants and to reflect on their own individual consumer behaviour in light of the needs of the natural world, other people, cultures and countries, and future generations.** | **3.4. The learner is able to envision sustainable lifestyles.** **3.5. The learner is able to feel responsible for the environmental and social impacts of their own individual behaviour as a producer or consumer.** | |

Objectives not promoted by the module in respective domains:

Cognitive domain/Holistic approach:

1.3. The learner knows the roles, rights and duties of different actors in production and consumption (media and advertising, enterprises, municipalities, legislation, consumers, etc.).

Conative domain/Achieving transformation:

2.2. The learner is able to evaluate, participate in and influence decision-making processes about acquisitions in the public sector.

Conative domain/Achieving transformation:

2.3. The learner is able to promote sustainable production patterns.

Social-emotional domain/Achieving transformation:

3.2. The learner is able to encourage others to engage in sustainable practices in consumption and production.

### 6.2.3. Competence-Oriented Learning in the Teacher Training Program

Primary education curricula are competence based and necessitate appropriate learning pedagogies. Gamification or learning by playing, for instance, increases the quality of learning by working at a physical level—it is kinaesthetic, rational and emotional, and is employed by working experientially and playfully [75]. The choice of contents is also important: contents related to climate change, for instance, are too abstract and complex for childhood and primary education pupils. Topics related more closely to everyday experience, such as waste, are recommended. Personal perspective plays a role—in this module, pre-service teachers are expected to be aware of their own consumer habits, values, the impacts and consequences of their consumer choices not only at local but also global level. Thus, promoting self-awareness and critical thinking at a personal level can lead to transformative practices.

From the point of view of professional practice, there is a need to break the perpetuation of established teaching practises that are insufficient for developing sustainability competences, and encourage competences related to building capacity for making change in different dimensions. However, in the study by Cebrian and Junyent [72] that is related to competences for ESD, the authors found that student teachers felt more secure in the process of acquiring the knowledge and practical skills related to nature and natural sciences than learning how to guide their future primary school students in the process of developing ethical values, managing their emotions and developing a positive relationship towards sustainability.

Although students in this module are encouraged to work on future scenarios (the ideal sustainable school), there is a lack of effective action. Actions that promote change towards improving the community or the environment are not integrated into the module, for instance, there is no use of service learning pedagogy, an approach supporting students' global skills development through experience in real-life situations beyond the university: service learning provides opportunities for learners to apply their classroom learning for the benefit of their communities [76] (p. 16). Effective actions related to the development of global competence [76] are also missing in the module.

### 6.3. ESD Professional Development of University Educators

### 6.3.1. Case Study Description

The literature is consistent in stressing the importance of the need to develop policies and institutional commitment to embed sustainability as a whole-institutional priority in higher education. Equally important is the need to develop the competences of university educators so that they are able to design and deliver curricula that helps students understand the sustainability challenge and address it through their personal and professional lives [77].

The Global Action Program (GAP) on ESD includes a priority goal related to the training of educators and researchers in ESD in order to address the SDGs more effectively [78]. However, in practice, there are only a few studies and initiatives that have reflected on how staff development in the area of ESD can influence change in higher education teaching and learning [34,79,80].

The 'University Educators for Sustainable Development' (UE4SD) was a project funded by the European Commission under the Life-Long Learning Programme—Erasmus Academic Networks and took place between 2013 and 2016. The project included 53 members (mostly higher education institutions) from 33 European projects. The main goal was to transform higher education teaching and learning towards sustainable development by enhancing the ESD professional development of university teaching and research staff. The partnership was unique in that it focused on the professional development of academic staff and the competences needed to transform university curriculum for sustainability. The project developed innovative professional development materials and selected best practice examples of programmes and courses at the European level. It also prompted discussions about the types and forms of professional development needed to embed ESD efficiently at the institutional and curriculum level, taking into account the different national and organisational realities. The

ultimate goal was to empower lecturers to understand and engage in ESD, focusing on the development of key competences to influence their own teaching practice and a wider organizational change.

The first stage of the project consisted of developing a state-of-the-art report that mapped out the situation in relation to ESD in higher education in Europe and showcased existing professional development programmes in ESD in country partners. The study highlights that ESD is becoming more prominent in higher education, but its key pedagogical principles and approaches are not usually known by university staff. The trend is to introduce specialist sustainability knowledge in certain places in the curriculum, rather than applying ESD pedagogy throughout the whole course of study. The report also analysed how professional development in the area of ESD was supported in the different participating countries. At a policy level, only a few countries recognize the need for training educators in ESD. In practice, there is also a lack of academic development in this area for university educators. In most cases, embedding ESD in the curricula is up to the individual interest and motivation of staff. The training that is offered mainly focuses on developing sustainable development knowledge and not on acquiring capabilities for influencing change in curriculum design and implementation. However, it also provides an overview of competences mentioned in the UNECE [47] document, supplemented by the country-specific competences in all domains [48] (pp. 45–49). This finding suggested universal applicability of this competence framework, as is further investigated here.

The second stage of the project sought to develop an online platform and a leading practice publication to feature good practice examples of ESD staff development initiatives for university educators across Europe. More specifically, the leading practice publication identifies and analyses 13 best practices, including examples of institutional programs, national initiatives and network activities. The authors of the publication highlight that although there are some interesting institutional initiatives in this area, key players in the development of staff competences have been ESD and higher education networks and associations [49].

### 6.3.2. Contents and Pedagogies Used

The UE4SD mapping exercise identified the type of ESD principles and approaches promoted in the ESD national policy documents of the project countries. Whilst many countries understand the need for participatory learning and systemic thinking to engage people in sustainable development, only a few promote critical and creative thinking and action learning. Future thinking was not mentioned in any of the policy documents analysed.

The results of the mapping study match the analysis of the 13 best practices on ESD professional development identified in the leading practice publication. The different initiatives analysed are underpinned by a wide variety of ESD pedagogical approaches, from participatory, collaborative and team-based learning methods, to action research and project-based learning, as a way to actively engage participants in critical and systems thinking and action taking [49]. Only a few of the initiatives stimulate creativity and the envisioning of alternative futures as pedagogical approaches to professional development. Systemic thinking and the whole-institutional approach are promoted in most programs, but only two of the thirteen specifically build leadership capabilities of educators to link ESD principles with specialist areas as well as to develop wider institutional practice in ESD. Mentoring is a key strategy used in many of the initiatives identified to ensure that individual competences are acquired and changes are influenced in the curriculum.

### 6.3.3. Competence Analysis

The design and delivery of a professional development program in the area of ESD should take into account the specific ESD competences that an educator must develop in order to shape the learning experience of learners so that they can grasp the sustainability challenge. Many universities currently engaged with the SDGs assume that it is enough to include contents related to SDGs in courses and modules. However, in order to implement the SDGs effectively, there is a need to question learning goals and influence change in how the curriculum is designed. Thus, staff development must be built

on critical reflection and participatory, action and transformative learning pedagogical strategies. The program should not be solely focused on the development of sustainability knowledge but should provide opportunities for educators to implement and evaluate key ESD competences [34].

In addition to this, it is crucial to understand that ESD competences are developed on a long-term basis and are best acquired when there is the involvement of a wide variety of stakeholders [34]. Acquiring transformative capabilities imply reflecting critically on the role of the educator; developing a transformative pedagogy that actively engages students in participation, creativity and future envisioning; and transforming education thinking, practice and systems [81]. Thus, educators must have the time to deeply engage in ESD thinking, experiment with new forms of teaching and engaging learners, and share their learning and experiences with other colleagues and stakeholders. Building competences related to system-wide and whole-institutional approaches will also require staff learning how to connect their practice with initiatives from the sector or institutional processes as well as working with stakeholders which may have not traditionally been involved during the shaping of academic programs.

The UE4SD project stressed the need to engage staff in developing ESD competences in order to mainstream sustainability in higher education. It demonstrated that there is a lack of quality professional development opportunities for educators to meaningfully engage in this area. Responding to this gap, the third and final stage of the project consisted of developing and piloting an Academy for ESD in Higher Education with the aim of developing the leadership capabilities of institutional teams in driving organizational change for sustainability.

## 7. Limitations of the Method

The research was built on qualitative research methods that provide a deep theoretical understanding of the problem which may underpin transformation in practice. In its initial stage, we identified two competence frameworks that are both validated and commonplace in scientific and policy discourse and are indicative of the "state of the art" of ESD debate. The former UNECE [47] framework for education for sustainable development uncovers differences in domains and dimensions of competences, providing thus an understanding of the role of competences in the learning processes. In this regard it has been used in previous efforts to evaluate diverse higher education programs on European level—and proved its applicability in a wide range of contexts (see also case study 3 in this article). The following academic debate concentrated on competences as learning objectives and their application on different levels, while these competence dimensions were not explicitly discussed in detail. This was the case of the latter UNESCO framework [3] that outlined key sustainability competences and further specified them as learning objectives for each of the SDGs providing thus practical guidance for educators in this field. This research attempted to systemize the perspective of learning domains and dimensions that are inherent in competences in order to shed light on the role of competences in general.

In the initial step, the UNECE framework [47] was used to develop categories covering normative (learning to know, learning to do, learning to live together, learning to be) and transformative (holistic approach, envisioning change, achieving transformation) competence dimensions. To analyse the overarching UNESCO competence framework [3], these categories served as theoretically based codes that helped to identify elements of competences that are relevant for the competence dimensions. These elements were embedded in the former UNECE [47] matrix which enabled comparison of both frameworks. The results of this step provided an answer to the question of whether all of these dimensions are still (implicitly) present in the key sustainability competences outlined in the more recent UNESCO framework, and how (in which combinations of elements). Applied qualitative analysis—an open coding method based on grounded theory—led to the development of a draft analytical tool that was tested in several steps described below. The limitation of this method (particularly the limited generalizability of the results [82] and hence replicability of the tool) should thus be overcome.

To understand competence dimensions across different contexts, principles of the conceptual framework analysis method [45,46] were used. Using a range of previous research results and literature sources, comparing them with emerging phenomena, and explaining this new understanding against the complex nature of competences, researchers outlined more a general analytical tool for analysing and possibly transforming these learning processes (where competences were set as educational goals). Use of both previous methods was based on the experience and understanding of the lead author; to avoid subjectivity in all of the steps, discussion between the authors was ongoing during the research.

As the conceptual framework analysis method combines arguments from different fields and discourses where the use of terms differs, there appeared to be difficulty with terminology. Thus the cognitive, conative and affective domain [4,5,54] may not fully overlap with cognitive, socio-emotional and behavioural domain [3], and there might also be a gap between these and Delors' [55] concept (learning to know, learning to do, learning to live together, learning to be). Each of these domains is positioned on a scale from systemic to personal (with the cognitive domain located at the systemic end of the scale, and increasing role of personal aspects further along the scale). Although the role of values was not examined here, it was supposed that normative knowledge (as opposed to instructional knowledge) played a different role in each of these domains. The 'transformative dimension' is also underpinned with values. We may hypothesize that envisioning in particular is a value-based process as it deals with identifying a desirable future. However, this aspect was not investigated further. The effort to unify the terminology should be not be considered as complete, and the role of values in this field, as inherent in sustainability science, should be further explored.

The case study method was used to test the framework. The selected cases were not representative of all HE practices and approaches; however, they depict different environments: a course for future experts; a teacher education module; and the situation in professional development of university educators. Although the structure of the case studies was similar, they were not comparable with each other but were useful in confronting theoretical findings with reality and illustrating pedagogical approaches.

## 8. Discussion and Conclusions

The presented research aimed to shed light on the distinction between systemic (instrumental) and personal (normative) aspects of competences in sustainability-oriented HE and reflect on them through the lens of transformation (and its underpinning competences). The point of departure was a theoretical comparison of cognitive, socio-emotional and behavioural learning domains identified by several authors (Table 1), which suggested the prevalence of systemic approaches (the cognitive domain was clearly distinguished and illuminated while the other domains are less clearly defined or even overlapping). If these domains are addressed separately, which is the case in the traditional model of teaching science, the values and related attitudes underpinning personal approach are at great risk of being put aside. In response, this research explored how to integrate all three domains in the context of HE, so that the traditional value-free model of scientific training can be challenged.

The authors questioned in particular how the normative dimension (covering systemic versus personal aspects) is inherent in sustainability competence frameworks relevant for HE level. They developed an analytical tool (see Figure 2), that was used to analyse: (i) the UNESCO key sustainability competence framework [3] (p. 10), and (ii) the SDG 12 learning objectives' framework [3] (p. 34) (see Tables 2 and 3, respectively). In the explored theoretical frameworks, the sustainability competences (or their elements) cover the analytical matrix more or less evenly thus showing an equilibrium between cognitive, socio-emotional and behavioural domains (systemic versus personal aspects in the normative dimension of these competences). Similarly, the general SDG 12 learning objectives were distributed also more or less evenly, while in the teacher education program in the University of Girona, the objectives in the transformative dimension were mostly not addressed (see Table 3 and its explanation). It is also of interest that none of the SDG 12 learning objectives that support transformation fall into the cognitive domain (most of them are in the conative domain). We may hypothesize that pure

scientific knowledge is not enough if used instrumentally to achieve the behavioural and social change outlined in this SDG, while working with the personal aspect of the normative dimension remains a challenging innovation in practice. This challenge was, however, addressed through the intervention competence introduced in the Mastertrack on Climate Change program, where students learn to connect abstract, scientific knowledge to their own experiences, and envisage and further perform transformational changes. It is important to note that 'while each competency has its own qualities and areas of relevance, they are mutually interdependent' [83] (p. 45).

Under certain circumstances, non-cognitive components of competences [20] may be considered as redundant for achieving professional expertise, and competences in this domain are thus considered as not worth developing. However, as is visible from Table 2, some of the key competences in the normative dimension (e.g., the emotional domain) are associated with 'envisioning change' and 'achieving transformation'. Omitting the emotional ('learning to be') domain would thus probably lead to restricted capacity for anticipating and realizing changes at personal and societal level (however, the interrelationship of intrinsic change processes with extrinsic transformation should be further explored). Similarly, focusing only on 'holistic thinking' in the transformative dimension (even if all domains in the normative dimension are addressed), learners would lack many capacities that are crucially needed for operating in the current, accelerating social, economic and environmental changes. Distinguishing learning domains and dimensions of competences thus not only shows the need for the balanced development of human capacities, it also may help to identify desirable learning processes that may be initiated through appropriate pedagogies (necessarily as diverse as the capacities they foster). The relevant didactic strategies have been described as having a pivotal role in developing sustainability values, skills and behaviour on HE level [27], and evaluation tools to assess the progress in this non-cognitive field have also been developed [84]. As the transformative dimension of learning is a novelty in the educational system in general (see above and case study 2), this aspect should be reflected not only in theory, but also in training educators on all educational levels to deal with this challenge in practice [34].

Regarding the interdependence between pedagogies needed for competence development discussed in this article, two contradicting practical consequences may be outlined: first, if HE institutions aim merely to teach instructional knowledge and sustainability competences are understood instrumentally [11], then no substantial change in teaching/learning approaches is required. However, we assert that sustainability competences inevitably include a *personal dimension* (values, emotions and motivation), as a condition for developing and applying normative knowledge and meeting learning objectives important for integrative competence development in the socio-emotional and behavioural domains. Consequently, pedagogical discourse should receive much more attention in the HE context, as demonstrated in the third presented case study. The relationship between competence-oriented teaching/learning and relevant pedagogies has also been suggested by other authors [57,85,86] but as the empirical findings do not support close interdependence [25], it is imperative that this be the subject of further research.

**Author Contributions:** J.D.—lead author, research design and method; R.H.—author of Case study 2; I.M.—author of Case study 3; F.P.S.—author of Case study 1; L.H.—literature review, editing and proofreading.

**Funding:** The research conveyed by the Charles University team was funded by Technology Agency CR, grant number TL01000117. The APC was funded by theCOPERNICUS Alliance.

**Acknowledgments:** The research team was established within the COPERNICUS Alliance network of higher education institutions, which made this publication possible. We thank Marie Pospíšilová for methodological advice, and Jiří Dlouhý who helped us with the figures and tables. We also thank the anonymous reviewers for their very useful comments.

**Conflicts of Interest:** The authors declare no conflict of interest.

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
