# Peer review of "Competences to Address SDGs in Higher Education—A Reflection on the Equilibrium between Systemic and Personal Approaches to Achieve Transformative Action"

_sustainability, doi:10.3390/su11133664_

Round 1

Reviewer 1 Report

In general, I  very much appreciate the work done in this paper and find the topic you are adressing of utmost importance! Especially I like the approach of finding an overview of various different models related to ESD competences  and raising awareness for emotional and behavioral competences in contrast to mere cognitve learning mechanisms.Feel encouraged to further work on this topic!

Yet, I believe that there should be major revisions especially concerning chapters 4-7 before publication, because the research method used is -in my oppionion- not sufficently explained and reflected: what about quality criteria of your research (objecitivity/intersubjectivity, reliability, validity?). How exactly are the "analytical tools" releated to each other and are (or are not) applied for the three case study descriptions? What's the difference between domains and dimensions of learning? In what way might the analytical tools be helpful for others? Etc. A more in depth description and discussion is necessary to help the reader follow your approach.

The case studies presented -in my point of view, are very interesting per se, but the subchaperts are inconsitent in their structure and not clear in their overall findings. Is there any chance to summarize and depict your findings in relation to questions posed in chapter 2? What exactly is the goal for bringing up the cases? Obviously not to test the applicability of the developed analytical tools, is it? Why, in case study 2, the "analytical tool" is applied for SDG 12, not for the educational program described?

Why were the models introduced in chapter 6 not integrated into the "analytical tools" presented in chapter 4?

Although I agree with the point made in chapter 7 I'd be careful if this point really can be termed an outcome and conclusion of your case study analysis.

Some futher recommendations: It might be helpful to point out from which disciplinary backgrounds exactly you do encounter the topic of the paper, because there might be (and are) very many different (disciplinary) approaches to (ESD) competences;

You are more than once taking about a "prevailing science model" in the paper - which model exactly do you mean by this?

Which relation is there between "normative - and transformative" on the one hand and "instrumental/systemic - normative/personal" on the other (this seems not clear to me)?

You might also check for idiographic-nomothetic distinction as categories to describe individual/specific vs. generalizing methods.

Table 2, in my oppinion, would be much more reader-friendly if abreviations would be at the very beginning ot the table (table 2 in any case is hard to read and not quite easy to follow in how it came to be).

Please also check for typos and some missings (e.g. in table 2 there seems to be a missing word related to "intervention competence="; see lines: 44; 87; 245-figure vs. table?; 339; 410;604;631;634)

Why there are crossed out paragraphs within table 3?

Might it be worth thinking about splitting the paper up into two? (One about the analytical tools developed and how they can be applied in gerneral and a second one dealing with the case studies and their relation to the analytical tools)?

Hope you find these comments helpful, please continue to work on this topic!

Author Response

Please find attached authors' responses.

Reviewer 2 Report

Dear Authors,

your paper is well structured and the faced issue is really interesting.

I suggest to improve the methodological section, by expanding the description of methods used for data collection and the connection between methodological framework and research questions.

Formal matter:

line 113 - different font

lines 232-233 - repeated lines (see lines 230-231)

line 282 - the abbraviation is lacking

table 3 - I suggest to explain also in the text the crossed out objectives

Finally, I think you could underline some practical implications of your research for HE institutions.

Author Response

(The authors gave the same response as above.)

Round 2

Reviewer 1 Report

Dear Authors,

the revised version of your paper reads to me as having clearly improved! It's much easier to follow your metodological approach and case studies now.

Besides some remarks and corrections (cf. commented paper version), to me still two major concerns remain which should be smoothed out for publication:

- Chapter 7 does, in my point of view, not yet clearly enough discuss and draw conclusions out of your reserach done; the methodological approach is not critically reflected at all. I suggest thouroughly rework on that and to state very clear which specific conclusions can be drawn because of the qualitative resarch you've conducted and the case studies you are presenting

- Introducing the idea of "intervention competence" seems rahter an add-on or seperate topic respectively than to be a coherent part of the paper. Especially case study 5.1. leaves the question open why it is necessary to introduce a "new" competence at all? This question also arises considering table 2. I get the impression that intervention competence is "squeezed in" somehow intstead of focusing on the topic of the paper (cf. title)

If there was a column in the journal explicitly for "approaches to discuss" or "tracking ideas within sustainability reserach", I would suggest to accept the paper after minor revisions now to open it up for discussion and further work on it;

But indeed, my impression is that in the given version the "technical soundness" and appropriateness of conclusions is not as clear as it should be in terms of a high standard journal publication yet. This might be a too critical perspective of mine since sustainability reserach is still an interdisciplinary endeavor and there are different disciplinary (methodological) standards. In order to make the paper reader-friendly for an interdisciplinary audience, I'd suggest another major revision.

Author Response

Dear editors and reviewers,

Thank you again for all your care given to the manuscript sustainability-484677. We have thoroughly considered all your comments and changed the manuscript accordingly. We also answer here more generally.

Comments and Suggestions for Authors

Dear Authors,

the revised version of your paper reads to me as having clearly improved! It's much easier to follow your methodological approach and case studies now. 

Thank you for appreciating our work!

Besides some remarks and corrections (cf. commented paper version), to me still two major concerns remain which should be smoothed out for publication:

Thank you for your precise work - all of the remarks have been addressed in the paper.

- Chapter 7 does, in my point of view, not yet clearly enough discuss and draw conclusions out of your research done; the methodological approach is not critically reflected at all. I suggest thoroughly rework on that and to state very clear which specific conclusions can be drawn because of the qualitative research you've conducted and the case studies you are presenting.

Thank you for pointing this out – the Chapter 7 has been carefully redrafted.

- Introducing the idea of "intervention competence" seems rather an add-on or separate topic respectively than to be a coherent part of the paper. Especially case study 5.1. leaves the question open why it is necessary to introduce a "new" competence at all? This question also arises considering table 2. I get the impression that intervention competence is "squeezed in" somehow instead of focusing on the topic of the paper (cf. title)

We have deleted the mentioning of the intervention competence at all places where it was considered to be redundant. As the essence of the research was thus not affected, this was also very useful guidance – thank you for that!

If there was a column in the journal explicitly for "approaches to discuss" or "tracking ideas within sustainability research", I would suggest to accept the paper after minor revisions now to open it up for discussion and further work on it; 

But indeed, my impression is that in the given version the "technical soundness" and appropriateness of conclusions is not as clear as it should be in terms of a high standard journal publication yet. This might be a too critical perspective of mine since sustainability research is still an interdisciplinary endeavour and there are different disciplinary (methodological) standards. In order to make the paper reader-friendly for an interdisciplinary audience, I'd suggest another major revision.

This requirement has provided an opportunity to substantially improve the article and we are grateful for that.

Round 3

Reviewer 1 Report

Dear authors,

appreciation for your sound rework of the method description and discussion chapters! From my point of view, the paper is now ready for publication in a high standard journal.

All the best.

Author Response

Dear reviewer,

thank you for your useful comments - we have tried to address all of them. Please see the detailed answer attached.

With kind regards

Jana Dlouhá

This manuscript is a resubmission of an earlier submission. The following is a list of the peer review reports and author responses from that submission.